# A Brief Review on the Potential of Psychedelics for Treating Alzheimer’s Disease and Related Depression

**DOI:** 10.3390/ijms241512513

**Published:** 2023-08-07

**Authors:** Alexander Pilozzi, Simmie Foster, David Mischoulon, Maurizio Fava, Xudong Huang

**Affiliations:** 1Neurochemistry Laboratory, Department of Psychiatry, Massachusetts General Hospital, Harvard Medical School, Charlestown, MA 02129, USA; 2Depression Clinical & Research Program, Department of Psychiatry, Massachusetts General Hospital, Harvard Medical School, Boston, MA 02114, USA

**Keywords:** Alzheimer’s disease, dementia, depression, psychedelics, psilocybin, LSD, mescaline, DMT, ketamine

## Abstract

Alzheimer’s disease (AD), the most common form of senile dementia, is poised to place an even greater societal and healthcare burden as the population ages. With few treatment options for the symptomatic relief of the disease and its unknown etiopathology, more research into AD is urgently needed. Psychedelic drugs target AD-related psychological pathology and symptoms such as depression. Using microdosing, psychedelic drugs may prove to help combat this devastating disease by eliciting psychiatric benefits via acting through various mechanisms of action such as serotonin and dopamine pathways. Herein, we review the studied benefits of a few psychedelic compounds that may show promise in treating AD and attenuating its related depressive symptoms. We used the listed keywords to search through PubMed for relevant preclinical, clinical research, and review articles. The putative mechanism of action (MOA) for psychedelics is that they act mainly as serotonin receptor agonists and induce potential beneficial effects for treating AD and related depression.

## 1. Introduction

As one’s age advances, it is common to see a decline in some regions of cognition. The encoding of memory, attention, and reasoning ability has declined with age [1,2,3]. However, it is often the case that one’s degree of cognitive impairment exceeds what is currently considered the norm. An impairment that exceeds this is, initially, a mild neurocognitive disorder (NCD). The manifestation of mild NCDs is heterogenous; there are various subtypes, such as amnestic mild cognitive impairment (aMCI) [4,5]. A mild NCD is considered to have progressed to the ‘major’ stage when it has begun to significantly impact the person’s ability to complete activities of daily living. Major NCDs often, but not always, follow mild NCDs; notably, specific subtypes such as aMCI are more prone to progress to significant disorders, such as Alzheimer’s disease (AD) [5].

As the leading cause of age-related dementia, AD accounts for approximately 60–80% of dementia cases. The underlying causes of AD are thought to be the formation of amyloid plaques composed of Aβ amyloid and neurofibrillary tangles (NFTs) composed of Tau protein. However, many other features, including neuroinflammation and blood–brain barrier (BBB) deficiencies, make it clear that the disease and its causes are likely heterogeneous. Due to the nature of the disease, it is difficult to diagnose; cognitive measures, as well as some physiological tests to rule out other causes of dementia, are typically used. In no small part due to the difficulties detecting the disease in its early stages, ideally, before impairment has occurred, no effective treatments or prevention (beyond risk reduction) are available to combat the disease [6]. Aducanumab is presently the only FDA-approved treatment for the causes of AD, targeting Aβ amyloid [7].

At present, there are a few pharmaceutical treatments that seek to improve some of the symptoms of the disease. These include cholinesterase inhibitors, which increase the synaptic availability of acetylcholine; the level of improvement offered by such drugs is varied but the side effects are generally mild, brought on in part by the drugs’ actions on the peripheral nervous system. At present, a single glutamate antagonist, memantine, is also approved and available, with similar efficacy to those targeting cholinesterase [8].

The psychoactive class of drugs is a broad category that encompasses numerous legal compounds, such as caffeine and nicotine, and presently illegal ones, such as heroin and ecstasy. Psychedelics are a subclass of psychoactive drugs; though typically described as “hallucinogens,” their effects extend beyond, and in many cases do not include, the induction of hallucinations. Hallucinations are generally defined as the sensory experience of something that does not exist, for example seeing entities or hearing voices that are not real. Psychedelic drugs may represent an alternative treatment for people with AD. Psilocybin, D-Lysergic acid diethylamide (LSD), dimethyltryptamine (DMT), mescaline, and ketamine are classic psychedelic drugs. Much of their study occurred in the early–mid 1900s, with interest in the substances being renewed in the past few decades as cultural and legal constraints were eased [9]. Microdosing, or administering psychedelics at a dose below the threshold of altered perception (and often well below that which causes hallucinations) on a schedule that includes multiple dosing sessions has also increased interest [10]. Based on their effects on the mind, they might alleviate cognitive and/or psychiatric symptoms of AD or mild NCDs. Research shows promise in terms of the potential psychiatric benefits of these drugs. Herein, we will review some of the mechanisms of action common to psychedelic drugs and their respective studied effects on AD.

## 2. Depression and Apathy in AD

Overall, around 50% of patients with AD exhibit depressive symptoms [11], though around 20–30% met the Diagnostic and Statistical Manual of Mental Disorders (DSM) criteria per past studies [12,13,14]. Whether AD increases suicide risk is unclear, though increases in suicidal ideation (SI) are noted, particularly in the period that follows diagnosis [15,16,17]. A small but significant percentage of individuals endorse SI upon receiving an AD diagnosis, particularly older, unmarried persons with limited support systems [18]. Patients with dementia who meet the DSM-IV criteria for ‘sad mood’ are more likely to experience behavioral problems and troubling thought patterns, including guilt and SI compared to those without [19]. Apathy, a common feature of AD, presents as a lack of interest or motivation when performing daily activities, experiencing new things, etc. Both apathy and depression are present in a few dementia cases [19] and, though depression (per DSM-IV criteria) is not stable over time in AD patients, apathy is [20]. Depression and apathy can be particularly troublesome in elderly persons with AD because they may neglect self-care due to apathy. This, and other associated behaviors and outbursts, may negatively affect a patient’s mood and exacerbate behavioral problems, as well as harm caregivers’ mental health [21]. Isolation and other social/environmental outcomes of AD are likely the source of considerable distress for patients with AD, exacerbating or inducing symptoms of depression; this aspect has been particularly troublesome during the restrictions of the COVID-19 pandemic [22].

Though the conditions and symptoms of AD could reasonably trigger depressive symptoms, there are significant relationships between neurodegeneration, cognitive impairment, and depression. Though they comprise only a tiny percentage (<1%) of AD cases, patients with autosomal dominant AD have significantly higher rates of suicidal ideation; this increase was independent of the person’s knowledge of their AD-related genetic mutation status. On examination of the brain tissue of AD patients who had depression and those who did not, the former exhibited more substantial Aβ amyloid plaque and NFT tangle burden than the latter [23]. The degree of cognitive impairment appears to be correlated with the incidence and severity of major depression in subjects with AD [24]. Treatment of depression is of considerable importance and traditional drugs, such as selective serotonin reuptake inhibitors (SSRIs), are often insufficient for this purpose. More high-quality studies that observe the longer term effects of treatment are necessary to evaluate the presence of significant differences in the cases of depression in AD-affected persons and the general population so that more tailored treatment plans might be developed [25].

## 3. Serotonin in AD

Psychedelics exert their effects primarily through mechanisms stemming from serotonin- 5-Hydroxytryptamine (5-HT) receptor activation [26]. Serotonin is relevant to AD pathology in multiple ways. There is evidence of considerable aberration in serotonin activity in AD cases [27,28]. The density and activity, but not expression, of the serotonin transport protein decrease with age in transgenic mouse models of AD [29].

In terms of serotonin’s influence on pathological features of AD, serotonin signaling appears to be relevant to Aβ amyloid levels; Cirrito et al. found that SSRIs reduced levels of Aβ amyloid in the interstitial fluid of AD transgenic mouse models. The Aβ amyloid burden of patients taking antidepressant medications (which include but are not limited to SSRIs) was lower than in untreated individuals. Such links between neurotransmitters and AD pathology may explain why depression is a significant risk factor for AD [30]. Agonists for various serotonin receptors also promise to reduce neuroinflammation [31]. Thus, treatment with the relevant SSRIs and agonists to reduce Aβ amyloid and neuroinflammation may be beneficial for AD patients.

Serotonin activity may be relevant to the cognitive impairment seen in AD and MCI. Indeed, the neuronal loss typically seen in AD affects regions of the brain with high serotonin receptor density, such as the hippocampus. Kepe et al. studied serotonin 1A receptor activity in the brains of patients with AD; the team found that binding potential 5-HT1A was significantly decreased in the hippocampus of patients with AD, along with the loss of volume characteristic to the brain region even in the early stages of AD [32]. Halliday et al. investigated the brain stems of patients with AD and age-matched controls, finding that serotonin-synthesizing neurons were selectively affected, correlating with the prevalence of NFTs. The 5-HT2A receptor levels are also diminished in multiple brain regions, such as the frontal and temporal cortices; the degree of loss in the latter region was found to be correlated with the rate of cognitive decline [33]. It should be noted that a decrease in 5-HT2A activity and receptor quantity is observed with age [34].

Neurodegeneration, measured by significant decreases in hippocampal volume, is a crucial feature of early AD [35]. Decreases in hippocampal volume (grey and white matter) are also seen in cases of depression and chronic stress, further establishing the link between the two disorders [36,37]. Serotonin-based treatments for depression, such as SSRIs and psychedelics, might promote neuroplasticity and recovery of these areas, though studies of hippocampal volume recovery in treated patients in depression remission and those not in remission show conflicting results [38,39,40].

The activity of serotonin receptors, specifically 5-HT2B, has been found to have relevance to neuroinflammation; 5-HT2B knockout mice exhibit more remarkable morphological changes in microglia, a neuroimmune component of the brain, and a longer time to recover from a neuroinflammatory insult (peripheral LPS injection) [41]. Deficiencies in the function of serotonin transporters are also noted in transgenic mouse models of AD; increases in neuroinflammatory markers, translocator protein (TSPO, 18kDa), and IL-1β were observed to precede a reduction in serotonin-transporter density and activity, as shown by Mataxas et al. [29]. TSPO, which is also known as peripheral benzodiazepine receptor (PBR), is a transmembrane protein located on the outer mitochondria membrane (OMM) and is mainly expressed in glial cells in the brain. In addition, a decrease in the maximum velocity of the transporter was also noted as levels of Aβ40, but not levels of Aβ42, increased.

Deficiencies of serotonin may also be blamed for increases in aggressive behavior and depression often seen in patients with AD [42,43]. The incidence of depression is more common in patients with AD and other dementias than in those without. This depression can negatively impact patients’ emotional and physical health because they may neglect self-care due to apathy. On the other hand, population studies indicate a lower prevalence of psychiatric disorders among individuals who use psychedelics [44].

Altered glucose metabolism is also a common feature of AD. Positron emission tomography (PET) scans of patients with AD reveal that glucose transport is abnormal in AD [45,46], though this aberration likely does not account for the clinical abnormalities found in AD; degradation of synapses, for example, will also cause a decrease in glucose metabolism [47]. Glucose transport is considered the limiting factor in how quickly a tissue can metabolize glucose. Glucose transporters are notably reduced along the BBB, even though reduced efficacy of the glucose transporter protein, due to mutations of a single allele, has considerable detrimental effects on brain health, development, and cognitive capacity. The impact of various psychoactive drugs on brain activity is through increases in glucose utilization and uptake [48]. Though research examining 5-HT receptor activity and glucose transport in the brain is scarce, the receptor is also found in skeletal muscle cells; when the receptor is activated there, it stimulates the recruitment of plasma-borne glucose transporters to the membrane, thereby stimulating rapid increases in glucose uptake [49].

## 4. Dopamine in AD

Though dysfunction of the dopaminergic system is more commonly seen as typical to Parkinson’s disease (PD) [50], there is evidence that the system has relevance to, or is at least disrupted by, AD as well. Overall, dopamine and dopamine receptors are significantly diminished in people with AD, relative to controls [51], and activity of the dopamine-converting enzyme dopamine-β-hydroxylase is also notably reduced [52]. However, the degree of dopaminergic disruption is not necessarily consistent across all cases of AD and is notably more significant in cases where the patient is exhibiting apathy and extrapyramidal (Parkinson’s-like) signs [53]. Extrapyramidal signs are associated with a worsened AD progression and mortality [54,55,56]. The presence and severity of the observed extrapyramidal signs in AD appear to be related to the patient’s degree of apathy [57,58]. Since the bulk of the present research suggests that extrapyramidal signs are highly relevant to dopaminergic dysfunction [57], the state of apathy is probably associated with this dysfunction as well. Indeed, hypoperfusion in brain areas with typically high dopaminergic transmission is associated with the presence and degree of apathy in AD patients. Agonists and antagonists of dopamine receptors increase or decrease, respectively, reward-seeking behaviors, highlighting dopamine’s importance to the brain’s reward system [57].

Dopamine transmission has many functions relevant to AD. Dopamine involves multiple systems/mechanisms of cognitive function and memory [59,60]. As part of the brain’s reward system, dopamine is a critical component of motivation and reinforcement learning [61,62,63]. Neuroinflammation and activation of astrocytes and microglia, commonly seen in AD cases, are reduced by dopamine [64,65,66]. Drugs that target acetylcholine, notably one of the few mechanisms of symptomatic AD treatment currently in use, notably increase the release of dopamine [67]. Dopamine may also inhibit the formation of fibrils and aggregations of amyloid beta [68]. Acetylcholine-targeting drugs notably increase dopamine release [67]; certain psychedelics, such as LSD, also affect the dopaminergic system [69].

## 5. Psychedelics for Treatment of AD and AD-Related Depression

As mentioned, psychedelic compounds may pose an alternative treatment for AD and related depression. The chemical structures of the psychedelics used as potential AD therapeutic agents in this review are shown in the following Figure 1.

Growing experimental evidence indicates that psychedelics may act as agonists for the serotonin receptor and dopamine receptor or the sigma-1 receptor (S1R) to induce beneficial effects for treating AD and AD-related depression.

### 5.1. Psilocybin

Psilocybin, the active compound of so-called “magic mushrooms,” was synthesized in its pure form as “Indocybin^®^” in the 1960s. It is typically taken orally and detectable levels of the drug can be found in plasma after 20–40 min on an empty stomach; the full effect of an 8–25 mg dose is typically felt between 70–90 min. It is converted to psilocin by the liver and excreted in the urine, with a half-life of around 74 min. As mentioned, psilocybin interacts with serotonin and not with dopamine receptors as other hallucinogens (such as LSD) do. Specifically, it interacts with the 1A, 1D, 2A, and 2C 5-HT receptor subtypes [70].

Recent work by Shao et al. indicates that psilocybin (in a 1 mg/kg dose) can increase the formation of dendritic spines, particularly in the frontal cortex, increasing density in a manner more pronounced in female than male mice [65,67]. Dendritic spine formation is a crucial facet of synaptic plasticity and autopsies from patients with AD indicate that dendritic spine density is significantly decreased, along with distorted and unattached spines [71].

Barett et al. explored the effects of psilocybin and dextromethorphan (DMX) on various cognitive domains in a placebo-controlled study, testing performance on the circular lights task, balance tasks, Penn Computerized Neurocognitive Battery, as well as the (MMSE). Participants were healthy adults between the ages of 22–43 years. Both psilocybin (1–3 mg/7 kg) and DMX (40 mg/7 kg) slowed tasks of motor praxis; tasks of working memory were slowed with psilocybin during DXM-impaired response inhibition in a dose-dependent manner. Overall MMSE scores were unaffected by either drug at any dose [72].

Psilocybin has facilitated research on how Serotonin 1A and 2A receptors influence working memory and attention; Carter et al. investigated the effects of psilocybin on tasks of spatial working memory and multiobject tracking; the drug impaired only multiobject tracking and correlations between performance on the two tasks were eliminated with psilocybin treatment. Pretreatment with ketanserin, a 5-HT2A antagonist, did not impact performance changes, suggesting that they are due to interaction with other receptors, presumably 5-HT1A [73].

A growing body of studies indicates that psilocybin has great potential in treating depression [74], particularly in cases resistant to traditional treatments [75]. Though more research is necessary, a recent double-blind study indicates at least comparable efficacy to traditional SSRIs [76]. When dealing with high doses of psilocybin, specific, nonsensory events predict treatment efficacy quite well; at high doses, psilocybin and other psychedelics are likely best used to facilitate psychotherapy, not necessarily as a self-administered pharmacological treatment [77]. Two studies utilized this strategy for patients with a life-threatening cancer diagnosis; in a double-blind study, acute-dose psilocybin reduced anxiety and depressive symptoms following a single session up to the 6 or 6.5-month follow-up point [78,79]. Such a tactic could also be employed in early-stage AD, though special considerations will be needed depending on the subject’s degree of cognitive impairment. The potential mechanism of action (MOA) of psilocybin is shown in the following Figure 2.

### 5.2. D-Lysergic Acid Diethylamide (LSD)

Family et al. assessed the pharmacokinetic and pharmacodynamic properties and the safety and tolerability of periodic low-dose LSD (totaling 10–20 μg) in healthy older adults. The drug was well tolerated, with no significant adverse events compared to placebo. Physical and psychiatric evaluation showed no significant abnormalities or concerning (e.g., suicidal) thoughts. An increase in mild–moderate headaches was noted in the LSD group relative to the placebo. Subjective vigilance reports based on Five Dimensional Altered States of Consciousness (5D-ASC) indicated that LSD had a detrimental effect, though the authors suggest that the study environment may contribute to this. However, the dosages reported in the study did not have any significant cognitive benefit for participants [80].

LSD’s primary mechanism of action, the activation of serotonin receptors, is in line with other treatments for psychiatric disorders such as depression. Bershad et al. examined the effect of microdoses (max. 26 μg) of LSD on healthy human subjects. An increase in systolic blood pressure was noted in participants at 13 and 26 μg but other physiological measures did not change significantly. Doses lower than 13 and 26 μg notably had no discernible impact on subjective measures of drug effect and vigor, though no significant cognitive benefits were found relative to placebo. It would seem that LSD’s effects are highly dose-dependent but it is safe to use and causes some behavioral/cognitive changes at low doses [81]. High doses (75 μg) have been found to improve mood but had disruptive effects on cognition and focus [82]; it is clear that dose is a substantial factor to consider, especially when dealing with elderly subjects.

Interestingly, a single acute dose (1 mg/kg) of LSD has also been found to influence the expression of many genes related to plasticity, glutamatergic signaling, and cytoskeletal development in rat brains [83]. Hallucinogens, such as LSD and other 5-HT2A receptor agonists, also induced transcription changes in genes relevant to certain behaviors [84]. Cini et al. experimented with human neurons and animal models, finding that synaptic plasticity, learning, and novelty preference were enhanced by d-LSD [85].

Speth et al. investigated how the act of mental time travel, which relates to an individual’s narrative self, is changed by a 75 μg dose of LSD; notably, only reference to the past was significantly diminished relative to placebo, while the present and future were unaffected [86].

LSD is currently a subject of interest in treating major depression as well. Research by Grof et al. indicated that LSD-aided psychotherapy alleviated the anxious and depressive symptoms of individuals diagnosed with cancer, based on pre- and post-treatment examination; there was no comparison to a placebo, however [87]. In a more recent study, Gasser et al. compared a single high-dose (200 μg) LSD-assisted psychotherapy to a single low/active-placebo LSD (20 μg). A follow-up after 12 months showed a host of benefits, including a reduction of anxiety and increased quality of life in over two-thirds of participants [88,89].

It should be noted, however, that because LSD metabolism involves the cytochrome P450 family, drugs that modulate these enzymes may trigger adverse events; additionally, certain antidepressants can increase response to LSD, likely due to shared interactions with dopamine and serotonin neurotransmission systems [90]. The potential MOA of LSD is shown in the following Figure 3.

### 5.3. Mescaline

Mescaline is a hallucinogen partially homologous to LSD, originally derived from the peyote cactus. It has notably been used as a baseline to which the effects of other psychedelics are compared; it is far weaker but can provide longer-lasting effects in full doses (200–400 mg) than other psychedelics such as LSD [91]. Though peyote is the most well-known source, it is found in other members of the Cactaceae family of plants and tends to be most concentrated in the bud, or photosynthetic stem, portion [92]. It is most commonly taken orally but other methods, such as insufflation, have been reported [93]. As with other psychedelics, its primary action is through the serotonin receptors, with an exceptionally high affinity towards the 5-HT2C receptor and, to a lesser extent, the 5-HT2A and 5-HT2B receptors. Orally ingested mescaline is absorbed in the gastrointestinal tract and has a half-life of around 6 h in humans; mescaline is converted to 3,4,5-tri methoxyphenyl acetic acid (TMPA). Its long-lasting effects may limit its efficacy in clinical settings, as even a few hours after the administration of sodium succinate, a relapse can occur and the effects of the drug resurge, as the drug outlasts the antidote [92]. Research on mescaline’s impact on cognition is somewhat scarce but long-term use (in the context of Native American religious rites) appears to cause no long-lasting cognitive or psychiatric problems [91]. No interactions between mescaline and a currently in-use clinical drug have been reported [90].

### 5.4. Dimethyltryptamine (DMT)

DMT is an alkaloid traditionally derived from the leaves of *P. virids* and has historically been a part of ceremonial rites in regions of South America, along with three other alkaloids derived from other plants that work in concert to amplify the effects of DMT [94]. A derivative of the amino acid tryptophan, it is synthesized in mammals’ tissues (including the brain), albeit at deficient levels; more research is necessary to determine concentrations throughout the brain and whether they might be psychologically/cognitively relevant [95].

DMT is typically administered by insufflation, with its psychoactive effects being significant and, in some cases, entirely diminished when administered orally as peripheral metabolism converts it to inactive metabolites. The effects of DMT are attenuated. DMT has the highest affinity for the 5-HT1A receptor, though it also binds to the 5-HT2 receptor family; its effect is greatly diminished by 5-HT1A receptor antagonists [96]. Binding to other serotonin receptors of the one, two, five, six, and seven subtypes has also been reported [97].

Notably, other sites of action of DMT include the S1R; S1R knockout mice do not exhibit the hypermobility that is seen in wild-type mice with functional S1Rs [98]. S1R receptors also appear to have an immunosuppressive role, stimulating the anti-inflammatory cytokine IL-10 and lowering the expression of interferon-gamma and TNF-α [99,100,101]. Many psychiatric and neurological diseases have an immune component; chronic neuroinflammation is one of the key features of AD. Agents that can combat this inflammation may be helpful for this reason [102].

Although various psychedelics/serotonin receptor agonists such as DMT can potentially increase neuroplasticity, Cameron et al. found that DMT on a microdosing schedule (many/several low doses at intermittent intervals) did not influence the expression of genes related to plasticity, while singular high doses did [103]. This highlights the importance of the dose for any treatment consideration. Small doses, which did not produce any behavioral or social changes in the rats, may not exhibit the longer-term benefits that combat neurodegeneration. The potential MOA of DMT is shown in the following Figure 4.

### 5.5. Ketamine

Ketamine, unlike other psychedelics, is in medical use, though typically in specialist and veterinary anesthesia where resuscitation equipment is not readily available. It is administered either through inhaled powder or intravenous or intramuscular injection; ketamine taken orally lacks the hallucinogenic effects as it is quickly metabolized [104]. It acts on various receptor types, including N-methyl-D-aspartate (NMDA) receptors, opioid receptors, monoaminergic receptors, muscarinic receptors, and other voltage-sensitive calcium ion channels.

The first study noting the antidepressant effect of ketamine was by Berman et al. who found, in a placebo-controlled study, scores of depression symptoms from the Hamilton Depression Rating Scale, though the sample size was small (n = 7) and the standard deviations of the observed decreases were quite large [105]. Later studies have conflicting results, though a recent meta-analysis supports an antidepressant effect overall. However, long-term and more rigorous studies are necessary [106]. Reports of side effects, such as anxiety, increased irritability, and aggression [107], may complicate AD-related treatment, given that these are already frequent features of AD.

Perhaps due to its nature as a sedative agent, ketamine has significant effects on cognition and might prove problematic in terms of AD or AD-related depression treatment. Ketamine has been found to diminish recall and recognition memory, as well as attention [108]. Different isomers of ketamine have been found to have variable effects but reductions are still observed in the time following injection; many of these effects were transient, returning to near baseline levels from 15 to 60 min [109]. A 2014 study by Morgan et al. noted that chronic ketamine use caused deficiencies in spatial memory, navigational memory, and other hippocampus-related disturbances [110]. Given that these areas are often already disturbed in AD [111], these long-term effects necessitate further investigation.

On the other hand, ketamine has been proposed to be an AD therapy [112] and a potential treatment for AD-related depression [113]. Indeed, the recent preclinical and clinical data are accumulating, supporting the neuroprotective, anti-inflammatory, and neurocognition-improving roles of ketamine in AD, as summarized in [113]. The potential MOA of ketamine for its beneficial effects on AD patients may be via its neuroprotective action on neurons, glial cells, and astrocytes, lowering neuroinflammatory cytokines, and antagonistic action on NMDA receptors [113].

## 6. Potential of Microdosing

Microdosing, typically described as the administration of psychedelics at a dose well below the threshold at which the hallucinogenic effects are incurred, has been a subject of increasing interest. Although singular small doses of hallucinogens appear to offer limited, if any, benefit, following a schedule of regular doses may prove beneficial while limiting the necessity for in-person therapy/guidance and avoiding the effects of full doses, such as the psychologically-challenging ‘bad trip’ [114]. An assessment of microdosing LSD on humans indicates that singular low doses of drugs such as psilocybin and LSD have little effect based on the present research. Thus, adopting a regular dose schedule may be beneficial and avoid potential problems observed with the whole psychedelic/hallucinogenic experience. LSD and psilocybin are the most commonly used psychedelics for self-medication microdosing, with a majority of surveyed persons noting that microdosing hallucinogens gave them improvements in depression (71.8%), anxiety (56.55%), focus (58.97%), and sociability (66.56%) [115]; other surveys indicate that perceived benefits and perceived challenges are often disparate between individuals [116]. Microdosing has also seen increasing interest and shows promise. However, more research is needed concerning long-term low-dose psilocybin or LSD treatment, particularly toward outcomes related to psychiatric disorders such as depression [117].

## 7. Conclusions

Psychedelic research has gained momentum over the past few years. Since serotonin and dopamine neurotransmission systems have considerable relevance to dementia, treatments that target these systems, including some psychedelic drugs, may have benefits. However, the research is still relatively new and, despite promising results, methods of therapy and dosages must be refined to avoid adverse health or psychological consequences, particularly for patients with AD. Microdosing may be the ideal method for administering psychedelics without the presence of trained personnel, but much more research is necessary in this area.

## Figures and Tables

**Figure 1 ijms-24-12513-f001:**
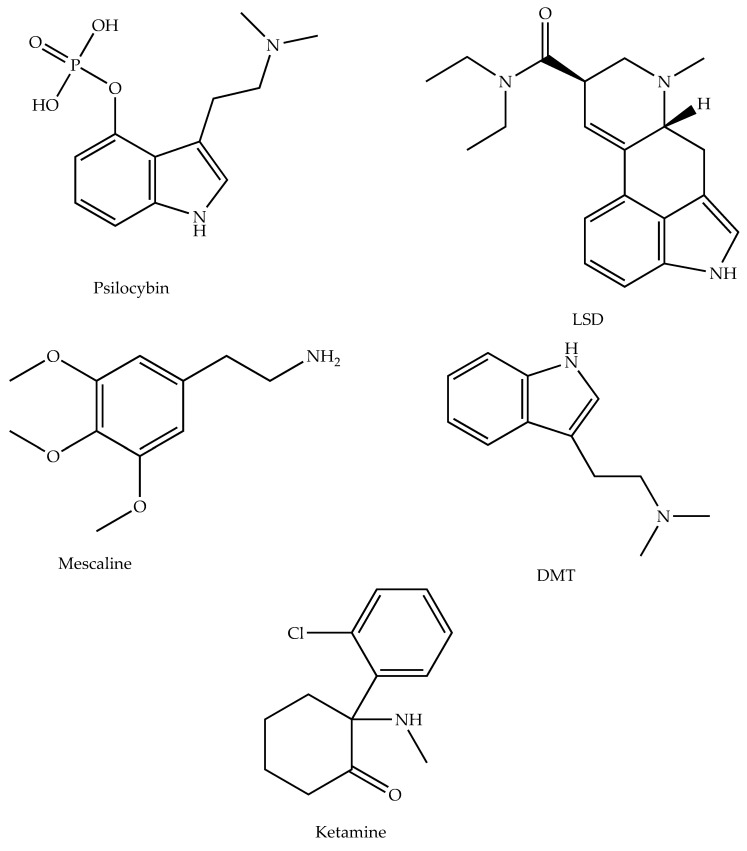
The chemical structures of psychedelics used as potential AD therapeutic agents—chemical structures created with ChemDraw.

**Figure 2 ijms-24-12513-f002:**
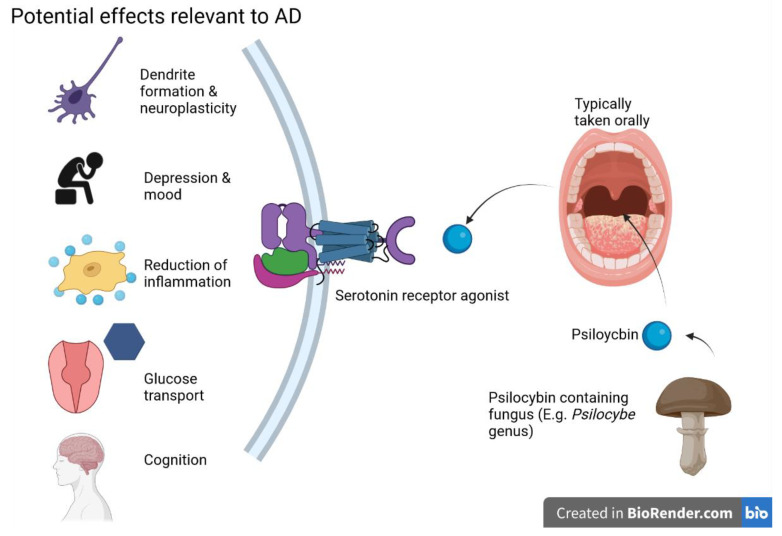
Psilocybin and its potential effects on AD are primarily exerted through serotonin receptor activity—figure created with Biorender.com (accessed on 19 June 2021).

**Figure 3 ijms-24-12513-f003:**
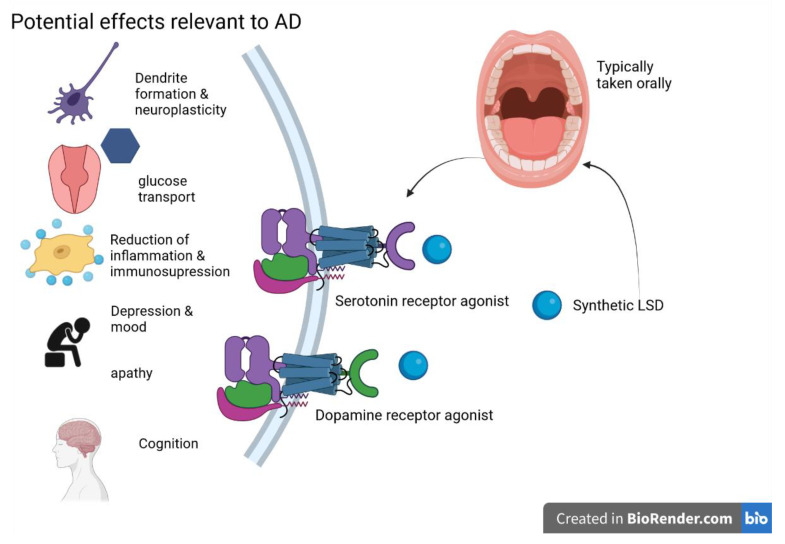
LSD and its potential effects on AD are primarily exerted through serotonin and dopamine receptor activity—figure created with Biorender.com (accessed on 19 June 2021).

**Figure 4 ijms-24-12513-f004:**
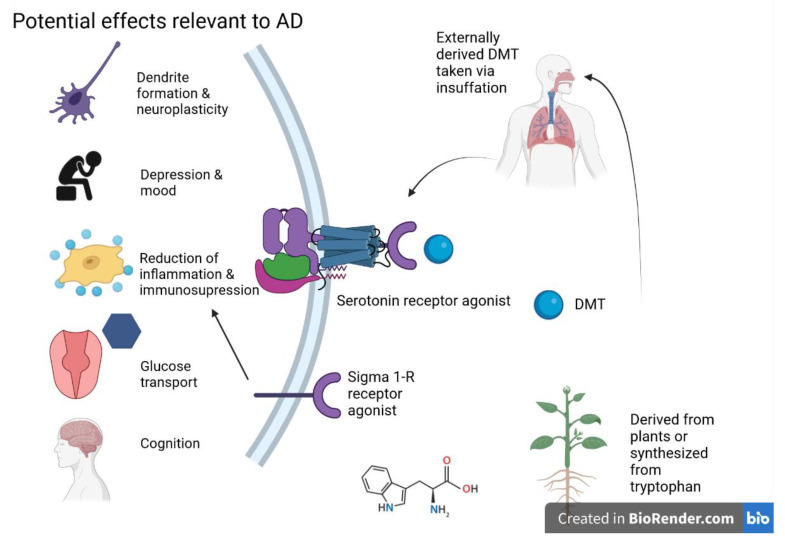
DMT and potential effects on AD are primarily exerted through serotonin and sigma 1-R receptor activity. Figure created with Biorender.com (accessed on 19 June 2021).

## Data Availability

Not applicable.

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
