# Peer review of "A Brief Review on the Potential of Psychedelics for Treating Alzheimer’s Disease and Related Depression"

_ijms, 2023, doi:10.3390/ijms241512513_

Round 1

Reviewer 1 Report

Reviewers' Comments to the Authors:

This in an interesting study that authors evaluated the potential of psychedelics for treating AD

and related depression in a narrative review.  There are however some issues that need to be addressed before this manuscript can be considered for publication (see my Comments below).

SPECIFIC COMMENTS TO THE AUTHORS:

1. Title:

Authors can mention in the title that his article is a review.

2- Abstract

-Which type of articles were included and the keywords can be mentioned.

-The main mechanisms can be mentioned. Which one can be used in clinical studies?

3- Introduction

-In this study, the goal of study was to review the articles for the effectiveness of psychedelics on AD and its related depressive symptoms. There are several reviews have been published recently focusing on using psychedelics on AD. The main superiority of this study is to investigate the effects of these drugs on AD-related depression. However, there was no literature to support this in the introduction. In addition, this should be explained in the last paragraph of introduction.

details.

4- Methods

-Which methods and keywords have been used to induce the articles to review? Which search engines have been used to find the related articles?  

5-Subtitles

- Depression and apathy in AD

-The molecular mechanisms of AD-related depression should be explained.

-Psychedelics can target AD and AD-related depression. Why serotonin and dopamine-related pathways were selected as main targets of psychedelics?

- Serotonin

In this part, authors should explain how serotonin pathways are affected in AD and AD-related depression. Then, the effects of psychedelics by targeting serotonin pathways should be explained in the related subtitle (The psychedelics as potential Alzheimer’s disease therapeutics).

-Dopamine in Alzheimer’s disease

There is no literature to support the involvement of dopamine pathways in the pathogenesis of AD-related  depression.

- The psychedelics as potential Alzheimer’s disease therapeutics

This subtitle should be changed as “Psychedelics for treatment of AD and AD-related depression “

There should be a paragraph to explain how psychedelics can generally target Dopamine and Serotonin in AD and AD-related depression.

6-Figures

Figure 2 to figure 4 can be integrated in one figure with focusing on AD and AD-related depression.

7-Tables

Authors can design a table to summarize the anti-AD and anti-depressant effects of potential psychedelics

8-References

-Recently published articles should be considered.

Author Response

Reviewer 1’s comments and authors’ responses

This in an interesting study that authors evaluated the potential of psychedelics for treating AD and related depression in a narrative review.  There are however some issues that need to be addressed before this manuscript can be considered for publication (see my Comments below).

SPECIFIC COMMENTS TO THE AUTHORS:

  1. Title:

Authors can mention in the title that his article is a review.

Response: Yes, the title has been changed to “A brief review on the potential of psychedelics for treating Alzheimer’s disease and related depression”.

2- Abstract

-Which type of articles were included and the keywords can be mentioned.

-The main mechanisms can be mentioned. Which one can be used in clinical studies?

Response: Yes, we have revised the abstract as suggested by the reviewer.

3- Introduction

-In this study, the goal of study was to review the articles for the effectiveness of psychedelics on AD and its related depressive symptoms. There are several reviews have been published recently focusing on using psychedelics on AD. The main superiority of this study is to investigate the effects of these drugs on AD-related depression. However, there was no literature to support this in the introduction. In addition, this should be explained in the last paragraph of introduction in details.

Response: We have elaborated beneficial effects of psychedelics on AD and its related depression in the sections after Introduction, citing many supportive references.

4- Methods

-Which methods and keywords have been used to induce the articles to review? Which search engines have been used to find the related articles?  

Response: As aforementioned, we used listed keywords and searched through PubMed for related articles used in this review.

5-Subtitles

- Depression and apathy in AD

-The molecular mechanisms of AD-related depression should be explained.

-Psychedelics can target AD and AD-related depression. Why serotonin and dopamine-related pathways were selected as main targets of psychedelics?

- Serotonin

In this part, authors should explain how serotonin pathways are affected in AD and AD-related depression. Then, the effects of psychedelics by targeting serotonin pathways should be explained in the related subtitle (The psychedelics as potential Alzheimer’s disease therapeutics).

-Dopamine in Alzheimer’s disease

There is no literature to support the involvement of dopamine pathways in the pathogenesis of AD-related  depression.

Response: Yes, in the relevant sections, we have explained putative molecular mechanisms of AD-related depression and its association with AD pathology. We have also given rationales why targeting serotonin, dopamine, and even the sigma-1 receptor system by psychedelics may be their putative mechanism of action (MOA) by citing many supportive references. Many literatures are cited in the manuscript, supporting the involvement of dopamine pathways in the pathogenesis of AD-related depression.

- The psychedelics as potential Alzheimer’s disease therapeutics

This subtitle should be changed as “Psychedelics for treatment of AD and AD-related depression “

There should be a paragraph to explain how psychedelics can generally target Dopamine and Serotonin in AD and AD-related depression.

Response: Yes, we have changed the subtitle and a short paragraph, as suggested by the reviewer.

6-Figures

Figure 2 to figure 4 can be integrated in one figure with focusing on AD and AD-related depression.

Response: Figures 2, 3, and 4 share the main MOA- serotonin receptor agonist. However, Figure 3 has an additional dopamine receptor agonist, and Figure 4 has other S1R agonist. We thus keep three different figures to highlight their different MOA.

7-Tables

Authors can design a table to summarize the anti-AD and anti-depressant effects of potential psychedelics

Response: The anti-AD and anti-depressant beneficial effects of potential psychedelics have already been summarized in the figures; a table may be redundant.

8-References

-Recently published articles should be considered.

Response: Yes, we have included some recently published articles, such as references 38, 41, 59, 66, 75, 77, 81, 114, 118, etc., published in 2020 and later.

Reviewer 2 Report

In this review the authors have made their efforts to comprehensively review the potential benefits of psychedelics in the management of AD and its related depression and apathy.

Please consider following comments 

1. Provide the reference for line 102-105.

2. line 112: is it inflammation or neuroinflammation? 

3. 112-114 is unclear. Please rewrite this sentence. For this claim provide the reference if available. 

4. line 191-192- Provide the reference 

5. line 212:  use AD abbreviation instead of Alzheimer's disease

Create the abbreviation section after conclusion part and include the following abbreviations with form (can also include other abbreviations which are elaborated in manuscript in their first appearance). This will help readers. 

PET

DSM

SSRI

MMSE

Author Response

Reviewer 2’s comments and authors’ responses

In this review the authors have made their efforts to comprehensively review the potential benefits of psychedelics in the management of AD and its related depression and apathy.

Please consider following comments 

  1. Provide the reference for line 102-105.

Response: Yes, we’ve provided the references in the text.

  1. line 112: is it inflammation or neuroinflammation? 

Response: Yes, we’ve changed it to neuroinflammation.

  1. 112-114 is unclear. Please rewrite this sentence. For this claim provide the reference if available. 

Response: Yes, we’ve rewritten this sentence for clarity.

  1. line 191-192- Provide the reference 

Response: Yes, we’ve added the reference

  1. line 212:  use AD abbreviation instead of Alzheimer's disease

Response: Yes, we’ve made the change.

Create the abbreviation section after conclusion part and include the following abbreviations with form (can also include other abbreviations which are elaborated in manuscript in their first appearance). This will help readers. 

PET

DSM

SSRI

MMSE

Response: Yes, we’ve added an abbreviation section in the manuscript.

Round 2

Reviewer 1 Report

 I am pleased to recommend accepting the article after minor revisions.

-Keywords should be capitalazied

Author Response

I am pleased to recommend accepting the article after minor revisions.

Response: Many thanks for the reviewer's encouragement and critiques.

-Keywords should be capitalized

Just to let the reviewer know, all the keywords have been capitalized as suggested.